# Does Technological Innovation Promote Green Development? A Case Study of the Yangtze River Economic Belt in China

**DOI:** 10.3390/ijerph18116111

**Published:** 2021-06-05

**Authors:** Senlin Hu, Gang Zeng, Xianzhong Cao, Huaxi Yuan, Bing Chen

**Affiliations:** 1Center for Modern Chinese City Studies, East China Normal University, Shanghai 200062, China; hsllh520@163.com (S.H.); cbing1995@163.com (B.C.); 2Economics School, Zhongnan University of Economics and Law, Wuhan 430073, China; huaxi@zuel.edu.cn

**Keywords:** technological innovation, green development, ecological modernization theory, Spatial Panel Durbin Model, panel threshold model, Yangtze River Economic Belt

## Abstract

The role of technological innovation (TI) in green development is controversial. Based on 2003–2017 panel data of 108 cities in the Yangtze River Economic Belt (YREB), this study constructed an index system to evaluate urban green development and analyzed the role of TI on urban green development with the help of a panel econometric model. The results show that: (1) From 2003 to 2017, the levels of TI and green development of cities in the YREB have gradually improved, but the core–periphery structure is obvious, and the levels of TI and green development in the lower reaches are significantly higher than those in the middle and upper reaches. (2) TI has a significant positive role in promoting green development, showing a U-shaped nonlinear relationship, and this relationship varies from region to region. (3) TI has a significant impact on green development with direct and indirect effects. In the economic and social dimensions, TI has a positive impact on green development, while in the ecological dimension, the direct effect and indirect effect have opposite relationships. (4) TI has a significant threshold effect on green development, and there are differences in threshold characteristics between the three dimensions. These findings provide a scientific basis for policymaking about innovation-driven regional green development, and it can enrich the related theories of environmental economic geography.

## 1. Introduction

The contradiction between economic development and environmental protection has aroused widespread concern globally [1]. Therefore, the United Nations has formulated 17 sustainable development goals and 169 specific goals (sub-goals) to form the cornerstone of the new 2030 agenda for sustainable development [2]. Over the past 40 years, China has achieved remarkable economic development, but this extensive pattern of economic growth at the cost of high energy input and high pollution output is unsustainable. China has become the world’s largest energy consumer, and fossil fuel energy still accounts for a large proportion of China’s total energy consumption [3,4]. As the biggest developing country in the world, exploring a feasible green development model will not only benefit Chinese people but also provide Chinese solutions for green growth to all countries.

The contradiction between development and environmental protection does not mean that the two pursuits are incompatible. Green development is considered to be an effective way to achieve a win-win situation between the economy and the environment [5,6]. The green development referred to in this paper is closely related to sustainable development. Its core emphasizes the promotion of economic and social development while protecting the environment, and it finally realizes the coordinated development of the economy, society, and environment. Green development is generally regarded as a means to achieve sustainable development in China [7].

Technological innovation (TI) is considered an important driving force for promoting regional green development. Scholars generally believe that TI is a source of economic growth [8]. However, with the increasing shortage of resources and the increasingly serious problem of environmental pollution, the research focus has gradually shifted to how to achieve TI-driven regional economic development in an environmentally friendly way [9,10]. According to the theory of ecological modernization, TI is one of the key factors to solving environmental problems [11,12]. Green development is a concept including ecological, economic, and social dimensions, which needs a comprehensive research perspective [1]. Generally speaking, regions are in different stages of development around the world, which leads to differences in the impact of TI on economic greening, ecological greening, and social greening. Given this, we took the Yangtze River Economic Belt (YREB) as a typical region, and empirically analyzed the following two problems: (1) Does TI affect green development, and is there spatial heterogeneity across the stages of economic development? (2) Can TI simultaneously promote economic greening, ecological greening, and social greening, and are there differences in the impact of TI on them? The purpose of this paper is to clarify the influence mechanism of TI on green development.

This study contributes to the literature in three ways. First, we constructed an evaluation index system of green development with 18 indicators and verified the nonlinear relationship between TI and urban green development from both direct and indirect perspectives. This finding supports the ecological modernization theory that green growth may be driven by TI. Second, we decomposed green development into three dimensions, namely economic greening, ecological greening, and social greening and then analyzed the effects of TI on them, which extended the previous research only on single dimensions or the overall situation. Third, the YREB was chosen as the research object to explore the spatial heterogeneity of the impact of TI on green development, which also extends the previous analysis of spatial heterogeneity in various countries.

The rest of this paper is organized as follows: Section 2 comprises the literature review and research framework, Section 3 presents the study area and data methods, and Section 4 features our main findings and discussion. Section 5 contains the conclusion and policy implications as well as the limitations and future research direction.

## 2. Literature Review and Research Framework

### 2.1. TI and Green Development

Although there have been many studies on TI and green development, relevant empirical studies are still inconclusive. They can be summarized according to three aspects:

TI facilitates green development [13,14]. First, TI can improve the efficiency of resource use, thereby reducing energy consumption and pollutant emissions per unit of output [9,15,16]. Second, TI can give rise to green industries and create new market demand [17]. Li et al. [18] found that innovation greatly promotes industrial green development. At the same time, other scholars believe that TI hinders green development due to the technological rebound effect [19,20]. Because of the opportunity cost of TI, to maximize profits, enterprises usually turn a blind eye to environmental problems and carry out TI based on saving labor and capital costs [10], which lead to resource consumption and environmental pollution. In addition, some studies confirmed that, due to the high initial cost of TI and the fact that the application of green TI will be limited by the level of economic development or other factors, the impact of TI on the green total factor productivity (GTFP) is not significant [21].

In recent years, an increasing number of studies came to the conclusion that there is a complex nonlinear relationship between TI and green development due to the multiple factors [22,23]. Gu et al. [24] confirmed the inverted U-shaped relationship between energy technological progress and carbon emissions and believed that such relationship existed with spatial heterogeneity. More importantly, TI is influenced by industry type, environmental regulations, economic development level, and other variables. For example, Omri [25] found that TI is only useful for environmental improvement in middle-income and high-income countries and that it has no significant impact on low-income countries. Wang et al. [23] believed that the impact of technological progress on carbon emissions varies according to the characteristics of the industry and proved, by using a panel quantile regression model, that technological progress in heavy industries can significantly reduce carbon dioxide emissions.

There is a vast body of literature on the relationship between TI and green development, but it mostly focuses on the relationship between TI and economic growth, energy consumption, carbon emissions, haze pollution, and other economic and ecological environmental factors [26,27] or the comprehensive empirical analysis of GTFP and ecological efficiency [14,28]. On the one hand, existing studies often ignored the impact of TI on social greening; on the other hand, few studies compared the effect of TI on the three pillars of green development (economy, society, and environment) from an empirical perspective under a unified research framework. A small number of studies have considered the three pillars of green development within an empirical analysis framework but mainly on the national scale and ignored the important role of the spatial spillover effect [25,29], which is also the problem solved by this research.

### 2.2. Research Framework

TI has a decisive influence on promoting economic greening, environmental greening, and social greening, and is very important for promoting sustainable development [30]. The research framework of TI on green development was preliminarily constructed from three dimensions: ecological greening, economic greening, and social greening (Figure 1). We think that ecological greening, economic greening, and social greening are interdependent, and economic development that ignores ecological and social development is non-green. Economic greening is an important embodiment of green development, ecological greening is the basis of green development, and social greening is the internal support of green development. Because of the complex spatial interaction effect, TI can not only directly promote or hinder the green development level (GDL) of a region by regulating the utilization rate of energy resources, energy consumption per unit GDP, and pollutant discharge, but can also be transmitted to the surrounding areas by indirect means such as technology spillover and competition [31,32], thus affecting the green development process of these areas.

China’s environmental pollution has seriously restricted the sustainable development of the economy and society [21,33]. It is particularly important to rely on TI to promote the development of ecological greening. In addition, China’s resource- and pollution-intensive industrialization has caused serious environmental pollution and deterioration [34]. Economic greening emphasizes the realization of high-quality economic development through industrial upgrading. TI can also increase social welfare and promote inclusive green growth through green consumption promotion [35,36].

## 3. Data and Methodology

### 3.1. Study Area and Data Sources

The YREB is one of the main axes of the T-shaped spatial development pattern of China [37], and also one of the major national strategies for China’s regional development in the new era (Figure 2). The study area includes nine provinces and two municipalities directly under the central government, namely Shanghai, Zhejiang, Jiangsu, Anhui, Jiangxi, Hubei, Hunan, Chongqing, Sichuan, Yunnan, and Guizhou.

There are two reasons for choosing the YREB as the research area: (1) It has an important strategic position for China, but the ecological environmental problems are prominent. The YREB runs through the three regions of East, West, and Central China. Its area is 2.1 million km^2^, accounting for 21% of China’s land area, but in 2018 the population density and economic density were respectively 4.5 times and 6.2 times the national averages. The YREB also is an ecologically fragile zone, and the situation of ecological environment protection is grim [38]. (2) The YREB is an ideal research area. On the one hand, because the differences between the lower, middle, and upper reaches of the YREB represent the development differences between the eastern, central, and western regions of China (the proportion of per capita GDP in 2018 was 2.41:1.11:1), it can be used to verify the spatial heterogeneity of the impact of TI on green development. On the other hand, due to the relative independence and system integrity of the sample area, it is the appropriate target for the spatial economics model [38].

Since there are no comparable statistical data on China’s Autonomous prefectures, we finally selected the data of 108 prefecture-level cities in the YREB from 2003 to 2017 as the research sample. The patent data used comes from the China’s State Intellectual Property Office (SIPO), and the index system data of green development evaluation are derived from the China City Statistical Yearbook, China Urban City Statistical Yearbook, and China Regional Economic Statistical Yearbook from 2004 to 2018. The missing data were interpolated with the averages of adjacent years.

### 3.2. Methodology

#### 3.2.1. Green Development Index Construction Based on Pressure-State-Response Model

Based on the connotation of green development and the sustainable development index system of the United Nations [3], we constructed three first-class indexes of ecology, economy, and society to evaluate the green development of the YREB. To further reflect the internal logic of the subsystem, we used the classic framework of “Pressure-State-Response” (PSR) proposed by the OECD [5,39] and applied it to the construction of second-class indicators.

For the selection of third-class indicators, we mostly collected similar indicator systems, extracted their evaluation indicators, and then selected the indicators most recognized by similar studies through comparative analysis [18,25]; in addition, we also referred to some domestic and foreign representative indicator systems at the scale of cities and above [40,41]. The evaluation index system is shown in Table 1.

#### 3.2.2. Methods, Variables, and Data

(1) Entropy index method

Compared with the commonly used methods in environmental impact assessment [42], such as analytic hierarchy process (AHP), data envelopment analysis (DEA), similarity ranking of ideal solutions, the entropy index method can avoid non-objectivity and deviation. The entropy index method is based on the variation degree of each index value to objectively assign its value [43]. The specific steps are as follows:

(a) Calculating the proportion of a single indicator of a city in the total of the indicators:(1)$$S_{i}=u_{i}/\sum_{i=1}^{n}u_{i}$$

(b) Calculating the entropy of evaluation index:(2)$$H_{i}=-k\sum_{i=1}^{n}S_{i}\ln(S_{i})$$
where *k* = 1/ln(*n*).

(c) Calculating the entropy weight of each index:(3)$$W_{i}=(1-H_{i})/\sum_{i=1}^{n}\left(1-H_{i}\right)$$

(d) Calculating the GDL of cities in different years:(4)$$GDL=\sum_{i=1}^{n}W_{i}B*100$$
where *x_i_* is the value of the *i*-th indicator; $u_{i}$, ${u_{i}}^{-}$ is the standardized values of the positive and negative indicators, respectively; and *GDL* is the *GDL* of each city.

According to Table 1, we first calculated the *GDL* scores in the three dimensions of economy, ecology, and society by using the entropy index method, and finally summed up the scores of these three pillars to get the total green development score.

(2) Econometric model and estimation method

The stochastic impacts by regression on population, affluence, and technology (STIRPAT) model is widely used in the field of environmental economy [44]. This model can not only identify the impacts of TI, population, and affluence on the environment, but also randomly expand other important factors according to the specific situation of different regions [45]. Therefore, we extended the STIRPAT model to study the impact of TI on green development:(5)$$GDL_{it}=a_{i}+\beta_{1}\ln TI_{it}+\beta_{2}{\left(\ln TI_{it}\right)}^{{}^{2}}+\beta_{3}\ln A_{it}+\beta_{4}\ln P_{it}+\varphi X_{it}+u_{i}+\delta_{t}+\varepsilon_{it}$$
where *GDL* denotes the level of green development; *T*, *A*, and *P* are *TI*, affluence, and population, respectively; and *X* is other control variables. *u_i_* represents individual fixed effect; $\delta_{t}$ represents the time fixed effect; and *ε_it_* is the random error term.

The spatial spillover effect of sustainable development is the key research focus of the future [16,46]. Considering the spatial interaction effect of green development between cities, we further incorporated spatial factors into the model construction. Spatial Panel Durbin Model (SPDM) can further decompose the calculation results into direct effect and indirect effect, and it is more accurate and reasonable to explain the regression coefficient from the aspects of direct effect and indirect effect [47]. The SPDM is as follows:(6)$$GDL_{it}=a_{i}+\rho \sum_{i=1}^{n}W_{ij}GDL_{it}+\varphi X_{it}+\theta \sum_{i=1}^{n}W_{ij}X_{it}+u_{i}+\delta_{t}+\varepsilon_{it}$$

In Equation (6), *ρ* is the spatial regression coefficient; *X_it_* includes several control variables; *i* and *j* are individual cities; and *W_ij_* represents the spatial weight matrix. Two kinds of spatial weights matrix were selected: the inverse distance geographic matrix (*W*_1*ij*_) and the economic geographic distance matrix (*W*_2*ij*_). The specific formulas are as follows:(7)$$W_{1}{}_{ij}=\left\{ \begin{array}{l} 1/{d_{ij}}^{2}\;\;\;(i\neq j)\\ 0\;\;\;\;\;\;\;\;\;\;\;\;(i=j) \end{array}\right.$$
(8)$$W_{2}{}_{ij}=\left\{ \begin{array}{l} GDP_{i}\cdot GDP_{j}/{d_{ij}}^{2}\;\;\;(i\neq j)\\ 0\;\;\;\;\;\;\;\;\;\;\;\;\;\;\;\;\;\;\;\;\;(i=j) \end{array}\right.$$
where *GDP* is the gross domestic product of each city and *d_ij_* is the geographical distance between *i* city and *j* city.

(3) Panel Threshold Regression Model

The Panel Threshold Regression Model (PTRM) proposed by Hansen [48] can not only accurately estimate the threshold value but also test the significance of “threshold value,” which can avoid the statistical error and regression error caused by subjective judgment to a certain extent. Therefore, we further used the panel threshold model to analyze the threshold characteristics of TI on green development:(9)$$\begin{array}{c} GDL_{it}=a_{i}+\beta_{11}\ln TI_{it}\cdot I(\ln TI_{it}\leq \lambda_{1})+\beta_{12}\ln TI_{it}\cdot I(\ln TI_{it}\leq \lambda_{2})+\cdots \\ +\beta_{1n}\ln TI_{it}\cdot I(\ln TI_{it}\leq \lambda_{n})+\theta \sum_{i=1}^{n}X_{it}+\varepsilon_{it} \end{array}$$
where $\lambda_{1}$, $\lambda_{2}$…$\lambda_{n}$ denote the threshold value to be estimated; *I*( ) indicates the index function; that is, when the conditions in parentheses are met, the value is 1; otherwise, the value is 0.

(4) Variable selection

The basis of model variable selection is as follows (Table 2)

(a) Technological innovation (*TI*). Technological innovation is an innovation aimed at creating new technology or based on scientific and technological knowledge. From the perspective of knowledge production, patents are the core of national or regional innovation resources, the most valuable part of economic value, and the source of TI. Many scholars in China and elsewhere use patent data to measure regional TI level [49,50]. In addition, compared with the number of patent authorizations, the number of patent applications is less constrained by the examination of the authorized institution and has more timeliness [51].

(b) Affluence (*A*). With the improvement of people’s income, the scale, level, and structure of consumption will change accordingly, which will affect the quality of the environment [31]. Specifically, the consumption growth brought about by the improvement of income level will inevitably lead to the expansion of the social production scale, which will lead to more consumption of resources and energy [2].

(c) Population size (*P*). Ehrlich and Holdren [52] argued that population size is one of the key factors that affect the environment. Population growth tends to mean that consumer demand continues to rise, leading to the expansion of production scale and a surge in energy and resources consumption [53]. It may also lead to an increase in environmental pollution.

(d) Industrial structure (*IS*). Pollution caused by industrial development has always been a worry for governments at all levels [54], and there is a U-shaped relationship between manufacturing agglomeration and green economic efficiency [53]. The development gap between different cities in the YREB is large and the less developed areas will still produce various serious forms of pollution in the process of playing economic catch-up. We used the proportion of gross industrial product in GDP to represent IS.

(e) Openness (*FDI*). The level of openness to the outside world can influence regional green development through technological spillover, the Pollution Haven effect, and in other ways [55]. The more open a region is to the outside world, the more likely it is that it will acquire internationally leading cutting-edge technology and knowledge, promote the transformation and upgrading of local industries, and improve the level of regional green development. However, the improvement of opening to the outside world will also bring disadvantages, such as the influx of foreign enterprises with serious pollution, resulting in the Pollution Haven phenomenon [56].

(f) Environmental regulation (*ER*). The Porter Hypothesis holds that reasonable ER can stimulate enterprises to improve their production, operation, and technology levels and optimize the efficiency of internal resource allocation. At the same time, it can maximize the ecological compensation effect of enterprises to offset the costs caused by compliance with regulations and thus improve the competitiveness of enterprises [57,58].

## 4. Empirical Results and Discussion

### 4.1. Spatio-Temporal Change of TI and Green Development

#### 4.1.1. Temporal Variation of TI and Green Development

The GDL of the YREB on average is relatively low but presents a slow ascending trend. Based on the entropy comprehensive index method, we calculated the GDL index of 108 prefecture-level cities in the YREB from 2003 to 2017. The average GDL index of the YREB from 2003 to 2017 rose from 21.9 to 26.4, showing a fluctuating upward trend (Figure 3).

The year 2008 was the low point of the green development of the YREB, scoring a value of 20.1. China had a rapid growth in economic development from 2003 to 2008 [34]. At this stage, there were problems such as extensive production mode and lack of awareness of environmental protection. After 2008, China, on the one hand, focused on economic growth, and, on the other hand, took steps to repair the damaged ecological environment and eliminate large energy consumption and serious environmental pollution of industrial enterprises continually; therefore, the GDL obviously improved. Specifically, in the three dimensions, the level of economic greening and the level of social greening fluctuated and increased, and their levels in 2017 were relatively close. The level of ecological greening is relatively low, but it has been showing a slowly rising trend, which indicates that the low level of ecological greening is still a significant problem in China.

The level of TI in the YREB shows a growing trend. The variation trend of IPAs from 2003 to 2017 can also be divided into two stages, showing an exponential growth trend on the whole. From 2003 to 2008, the number of patent applications grew slowly. After 2008, the number of patent applications grew rapidly, reaching 63.6 per 10,000 of the population in 2017. The number of patent applications in the lower reaches, the middle reaches, and the upper reaches decreased in turn, among which the number of patent applications in the lower reaches reached 34.6, accounting for more than half of the total. The lower reaches of the YREB constitute the largest economic zone in China, as well as the most dynamic and promising one [59].

#### 4.1.2. Spatial Variation of TI and Green Development

The level of TI and green development of major cities in the YREB from 2003 to 2017 has increased significantly, but the overall trend is a step-down pattern of “lower, middle, and upper reaches” (Figure 4). In 2003, the number of cities with green development at medium and high level or above was relatively small and mainly distributed in the lower reaches. By 2017, the GDL of most cities had improved, and the cities with higher GDL in the lower reaches showed a Z-shaped distribution trend. Moreover, due to the advantages of the upper and middle reaches being less developed, the level of urban green development has improved significantly.

In 2003, patent applications were mainly concentrated in the central cities, presenting a point-like discrete distribution. In 2017, the number and scale of patent applications increased rapidly, being mainly concentrated in the lower reaches and the core cities in the upper and middle reaches, and the overall pattern changed from a discrete cluster pattern to a continuous cluster pattern.

### 4.2. The Influence of TI on Green Development

#### 4.2.1. Base Model

(1) Selection of Spatial Econometric Model

To ensure scientificity, we adopted the Lagrange multiplier (LM), likelihood ratio (LR), and Hausman methods to select the specific form of spatial econometric model (Table 3). First, both the LM(lag) test and the LM(error) test rejected the null hypothesis, indicating that spatial variables need to be introduced to explore the relationship between TI and green development. Second, LR_spatial_lag and LR_spatial_error also rejected the null hypothesis, indicating that the SPDM cannot be downgraded to the Spatial Panel Lag Model (SPLM) or the Spatial Panel Error Model (SPEM). Finally, the Hausman test results rejected the null hypothesis, so the fixed-effect (FE) approach should be used in the discussion of the relationship between TI and green development. The fixed-effect model can eliminate inter-individual heterogeneity and minimize endogenous bias due to missing variables.

(2) The Overall Effect of TI on Green Development

The regression results of TI in the YREB on the overall and sub-regional GDL are shown in Table 4. Technological innovation and quadratic terms of TI variables were added into the model, and all control variables were added at the same time. The empirical results show that: (1) In the models without the quadratic term of lnTI, the coefficients of variable lnTI in Model 1, Model 3, and Model 5 are significantly positive, that is, TI can promote urban green development in a direct or indirect way. (2) After adding the quadratic term of lnTI, the regression coefficients of (LnTI)^2^ in Model 2, Model 4, and Model 6 were all positively significant at the 1% level, which were 0.08, 0.038, and 0.045, respectively. This indicates that under the two weights W1 and W2, there is a steady nonlinear U-shaped relationship between the promoting effects of TI and urban green development. In other words, when the TI ability is low, the improvement of TI ability will hinder the improvement of urban GDL, but when the TI ability has crossed the inflection point, TI can positively promote the improvement of GDL.

TI mainly influences urban green development through direct and indirect ways. Further, the influence effects of the explanatory variables and control variables were decomposed [47] to obtain the direct and indirect influence effects of each variable on green development (Table 5). The results show that the impact of TI on the green development of the whole sub-region of the YREB in the research period is through both direct and indirect effects and is significantly positive at the 1% confidence level under the W1 and W2. This indicates that the improvement of a city’s TI level has a significant U-shaped nonlinear relationship with the green development of both local and neighboring cities. This is consistent with the findings of Omri [25] and Liu et al. [60].

In terms of control variables, each variable has an effect on the level of urban green development mainly through a direct effect. Population has an inhibiting effect on the improvement of GDL, while per capita GDP can significantly improve the GDL of cities. This is because population growth will lead to higher consumer demand and a surge in energy and resource consumption [53]. At the same time, with the improvement of the level of economic development, the public’s environmental awareness has been improved, and high-quality fuels have been popularized [16]. In addition, FDI, IS, and EI have a significant effect on the dependent variable. The FDI and ER variables can promote the improvement of urban green development through the direct effect. The main reason is that FDI and enhancement of ER can improve the TI capacity of cities and thus promote the development of the city’s green transformation development [21]. An increase in the urban industrial proportion will inhibit urban green development. Wang et al. [14] found that every 1% increase in the industrial proportion will reduce GTFP by 0.091%.

#### 4.2.2. The Influence of TI on Green Development in Three Dimensions

TI can affect three dimensions of urban green development, and all of them have significant nonlinear relationships (Table 6). Specifically, in the economic and social dimensions, TI has a U-shaped nonlinear relationship with green development, and both of them act on urban green development through direct and indirect effects. In the ecological dimension, the total effect of TI on green development is not significant; the main reason is that the direct effect and indirect effect have opposite nonlinear relationships. That is, the local TI has a U-shaped relationship on its green development, while the local TI has an inverted U-shaped relationship on the green development of neighboring cities. When the TI ability of a city is strong, it can play a demonstration role for the surrounding cities. However, when the TI ability of the city is improved to become the core city, it will often have a strong resource competition effect with the surrounding cities, and it may also transfer the pollution-intensive industries to the surrounding lesser developed cities [34].

In terms of control variables, the expansion of population and the proportion of secondary industry will directly or indirectly hinder the improvement of urban economy, ecology, and social greening level. However, FDI and ER have significantly promoted the greening of the economy.

#### 4.2.3. Threshold of TI on Green Development

Based on the PTRM, we found that TI has a single threshold effect on green development (Table 7). The threshold value of lnTI on the level of green development is 9.79 (17,854 units), and the coefficient of TI on the level of green development is 0.352. When the threshold is crossed, the coefficient increases to 0.708. This indicates that the improvement of TI ability can accelerate the promotion of urban GDL.

Specific to the three dimensions of green development, there is a single threshold effect in the economic and social dimensions, while there is a double threshold effect in the environmental dimension, and urban TI is more sensitive to the threshold of environmental greening. The thresholds of lnTI’s impact on the level of economic and social greening were 8.84 and 9.17, respectively. When the thresholds were crossed, the coefficients of lnTI increased significantly. The relationship between lnTI and ecological greening is complex, and there are double threshold eigenvalues, whose coefficients first increase and then decrease. This shows that the governance of ecological environment can not only rely on TI, but also need to introduce more measures for collaborative governance.

#### 4.2.4. Robustness Tests

To further explore the spatial heterogeneity of the impact of TI on green development and to ensure the reliability of the empirical results, we did the following robustness tests:

(1) Spatial heterogeneity analysis of the three sub-regions

The development of regions in the middle and lower reaches of the YREB are quite different, so the relationship between TI and urban green development in different regions can be used as the robustness test of this study. The empirical results (Table A1) show that there is a significant nonlinear relationship between TI and green development in the upper, middle, and lower reaches of the region, but the relationship curves are different in different regions. There is an insignificant U-shaped relationship between the total effect of urban TI capability in the upper reaches and green development, but there is an inverted U-shaped relationship between the total effect of urban TI capability in the lower and middle reaches. This finding supports the view of Omri [25], regarding the urban scale, that the impact of TI on the three dimensions will vary according to the national economic development stage.

(2) Replace the dependent variable with the number of invention patent authorizations in the city

Invention patent authorization refers to the valid patents approved by the China National Intellectual Property Administration, which directly reflects the achievements and ability of TI. According to our SPDW model analysis (Table A2), the U-shaped nonlinear relationship between TI and urban green development still exists. In the three dimensions, the relationship between TI and economic greening and social greening is still U-shaped, while the relationship between TI and ecological greening is inverted U-shaped. This is consistent with the empirical results based on the IPAs as the dependent variable, and also fully demonstrates the validity and credibility of the empirical results of this study.

## 5. Conclusions and Policy Implications

### 5.1. Conclusions

This paper decomposed green development into three dimensions, namely economic greening, ecological greening, and social greening and then analyzed the effects of TI on them, which extended the previous research only on single dimensions or the overall situation. At the same time, it can be said that this study is a further extension of the studies of Omri [25] and Brandão et al. [29] from spatial scale. The main conclusions of this study are as follows:

First, the level of TI and green development in the YREB is constantly rising from 2003 to 2017. Due to economic development, policy and market, the level of TI and green development in the lower reaches of the YREB is significantly higher than that in the middle and upper reaches, and that in the middle reaches of the YREB is significantly higher than that in the upper reaches.

Second, there is an U-shaped nonlinear relationship between TI and green development of cities in the YREB, which supports the viewpoint of ecological modernization theory. However, this relationship is significantly different in the upper, middle and lower reaches in the YREB. For example, in the lower and middle reaches, there is a significant inverted U-shaped relationship, while in the upper reaches, there is a U-shaped relationship.

Third, TI can have a significant impact on urban green development through direct or indirect effects and has an impact on the three dimensions of ecological greening, economic greening, and social greening. In the economic and social dimensions, the direct and indirect effects of TI on green development have a U-shaped nonlinear relationship. However, in the ecological dimension, the total effect of TI on green development is not significant because of the opposite relationships between direct effect and indirect effect.

Fourth, TI has a significant threshold effect on the promotion of green development, but the threshold of ecological greening is the lowest. There is a single threshold in economic and social dimensions, that is, after the threshold is crossed, the promoting effect will increase significantly. There is a double threshold in the ecological dimension, and the promoting effect increases first and then decreases, that is, the appropriate TI ability can promote the ecological greening better.

### 5.2. Policy Implications and Research Prospect

(1) Continue to increase investment in technological innovation and attach importance to the threshold effect of technological innovation.

According to the results of this study, the number of TI of most cities in the YREB has crossed the inflection point of the U-shaped curve, that is, the stronger the ability of TI, the more it can promote the improvement of urban GDL. In the future development of the YREB, the ability of TI should be further enhanced. When a city crosses the threshold of TI (17,854 IPAs), it will more effectively promote the improvement of the level of green development. From the perspective of spatial heterogeneity, the upper reaches should continue to strengthen its TI capacity. For the lower and middle reaches, TI is currently inhibiting green development, so regulation, IS optimization, and other ways should be used to synergize while enhancing technological capabilities. In a word, ‘carrots’ and ‘sticks’ are both necessary in the process of urban green development [61].

(2) Strengthen the synergistic effect between cities and coordinate the efforts from the three dimensions of ecological greening, economic greening, and social greening simultaneously.

The results show that local TI has a U-shaped relationship with the greening of neighboring cities, indicating that local TI has a positive spillover effect on neighboring cities. Specific to the three dimensions, local TI currently has an inhibiting effect on the green development of neighboring cities. In the future, the phenomenon of “beggar-thy-neighbor” between cities should be combated and the coordination of environmental governance between cities should be strengthened. This study scientifically constructed an evaluation index system of green development from the three dimensions of ecology, economy, and society, and verified that TI can affect the process of these three dimensions simultaneously through direct and indirect effects at the urban scale and that this relationship varies with the stage of economic development. These findings provide a scientific basis for policymaking about innovation-driven regional green development, and it can enrich the related theories of environmental economic geography. However, this paper also has some limitations. First, there is a complex interactive relationship between the three dimensions of green development, empirical studies can consider taking this factor into consideration. Second, not all technological progress can reduce the demand for resources [62], and TI can be divided into green TI and non-green TI, green TI can maximize the effective use of relevant resources [58,63], and the heterogeneous impact of green TI and non-green TI on green development can be further examined in the future.

## Figures and Tables

**Figure 1 ijerph-18-06111-f001:**
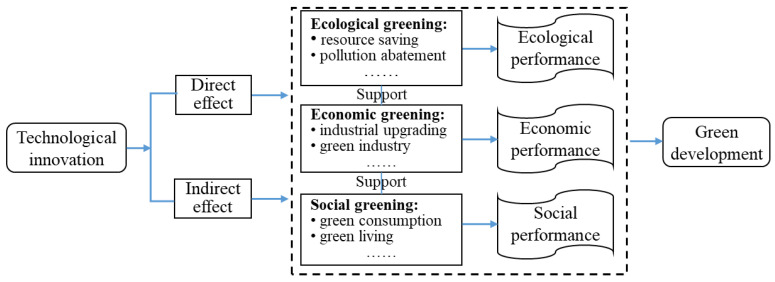
Research framework of the effect of Technological Innovation on green development.

**Figure 2 ijerph-18-06111-f002:**
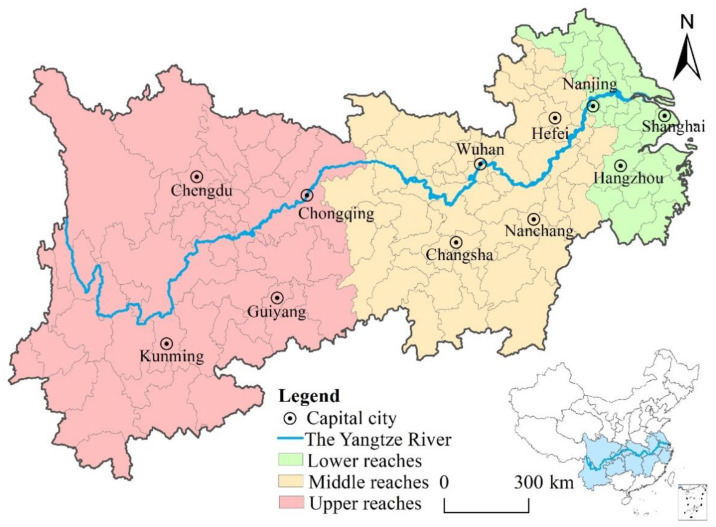
Location of the Yangtze River Economic Belt in China.

**Figure 3 ijerph-18-06111-f003:**
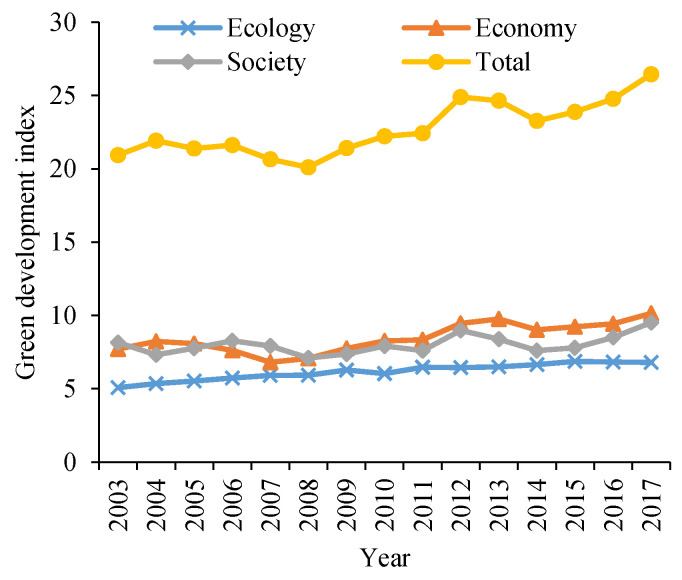
Evolution characteristics of urban technological innovation capability and GDL from 2003 to 2017. IPAs: invention patent applications.

**Figure 4 ijerph-18-06111-f004:**
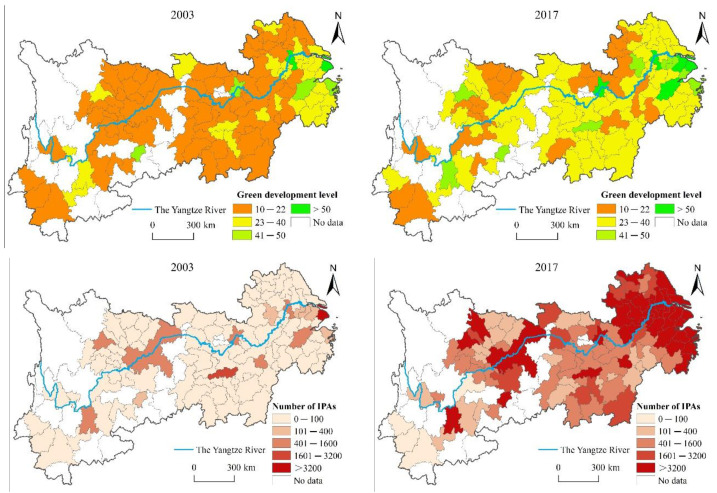
Spatio-temporal pattern of technological innovation and green development in the YREB in 2003, 2017. IPAs: invention patent applications.

**Table 1 ijerph-18-06111-t001:** Evaluation index system of green development.

Primary Indexes	Second-Class Indexes	Third-Class Indexes	Attribute
Environment	P	Industrial wastewater emission intensity (ton per km^2^)	Negative
		Industrial sulfur dioxide emission intensity (ton per km^2^)	Negative
	S	Green covered rate of urban built-up area (%)	Positive
		Park green area per capita (m^2^)	Positive
	R	Ratio of industrial solid wastes utilized (%)	Positive
		Harmless treatment rate of domestic waste (%)	Positive
		Ratio of industrial dust removal (%)	Positive
Economy	P	GDP growth rate (%)	Positive
		Average profit of industrial enterprises (yuan)	Positive
	S	Proportion of tertiary industry (%)	Positive
		Advanced degree of industrial structure	Positive
	R	GDP per unit of industrial electricity consumption (yuan per KWh)	Positive
		GDP per unit of industrial water consumption (yuan per ton)	Positive
		GDP per unit of urban construction land (yuan per KWh)	Positive
Society	P	Natural population growth rate (‰)	Negative
		Registered urban unemployment rate (%)	Negative
	S	Domestic water consumption per capita (ton per person)	Positive
		Domestic electricity consumption per capita (KWh per person)	Positive
	R	Number of doctors per unit (unit/10,000)	Positive
		Number of buses per unit (unit/10,000)	Positive
		Proportion of social security expenditure in government public budget expenditure (%)	Positive

Notes: P: Pressure; S: State; R: Response.

**Table 2 ijerph-18-06111-t002:** Descriptive statistics of variables.

Theme	Index	Calculation Method	Mean	Std. Dev	Min	Max	Obs.
Dependent variable	Green development level (GDL)	Calculated by Entropy index method	23.01	9.685	10.22	68.75	1620
Explanatory variable	Technological innovation (lnTI)	Number of invention patent applications (units)	5.197	2.240	0	10.93	1620
Control variables	Population size (lnP)	Resident population at year-end (ten thousand persons)	5.970	0.620	4.257	8.031	1620
	Affluence (lnA)	Per capita GDP (yuan)	10.05	0.850	7.771	12.00	1620
	Industrial structure (lnIS)	Ratio of output value of secondary industry in GDP (%)	9.747	2.013	0	14.43	1620
	Openness (lnFDI)	Proportion of foreign direct investment in GDP (%)	3.851	0.211	2.875	4.329	1620
	Environmental regulation (lnER)	Proportion of environmental protection investment in GDP (%)	0.410	0.436	0	4.128	1620

**Table 3 ijerph-18-06111-t003:** Identification test of spatial panel econometrics model.

Test	W1	W2
LM(lag) test	60.98 ***	9.686 ***
Robust LM(lag) test	3.202 *	0.565
LM(error) test	557.42 ***	158.02 ***
Robust LM(error) test	499.64 ***	148.90 ***
LR_spatial_lag	55.89 ***	40.37 ***
LR_spatial_error	68.24 ***	54.55 ***
Hausman test	133.36 ***	117.68 ***

Notes: Standard errors in parentheses; * *p* < 0.10, *** *p* < 0.01.

**Table 4 ijerph-18-06111-t004:** Overall regression results.

Variable	(1)	(2)	(3)	(4)	(5)	(6)
	FE	FE	SPDM (W1)	SPDM (W1)	SPDM (W2)	SPDM (W2)
lnTI	0.422 *** (0.160)	−0.460 (0.279)	−0.011 (0.099)	−0.43 *** (0.168)	0.022 (0.099)	−0.435 *** (0.169)
(lnTI)^2^		0.080 *** (0.025)		0.038 *** (0.014)		0.045 *** (0.014)
lnP	−0.872 (0.55)	−0.932 * (0.553)	−0.715 *** (0.276)	−0.756 *** (0.274)	−0.863 *** (0.276)	−0.882 *** (0.275)
lnA	2.149 *** (0.507)	2.193 *** (0.512)	1.697 *** (0.517)	2.044 *** (0.530)	1.192 ** (0.511)	1.511 *** (0.524)
lnFDI	0.009 (0.185)	0.017 (0.185)	0.194 ** (0.085)	0.203 ** (0.084)	0.182 ** (0.085)	0.184 ** (0.084)
lnIS	−6.235 *** (1.348)	−4.346 *** (1.452)	−2.294 *** (0.644)	−1.460 ** (0.686)	−2.313 *** (0.650)	−1.422 ** (0.695)
ER	0.491 ** (0.205)	0.551 *** (0.209)	0.442 *** (0.161)	0.439 *** (0.160)	0.381 ** (0.162)	0.404 ** (0.161)
W*lnTI				−1.835 *** (0.602)	0.649 *** (0.250)	−0.652 (0.612)
W*(lnTI)^2^				0.186 *** (0.044)		0.085 ** (0.040)
W*lnP				−1.087 (1.100)	−3.571 *** (1.065)	−3.933 *** (1.066)
W*lnA				−0.531 (0.837)	−1.098 (0.786)	−0.539 (0.864)
W*lnFDI				−0.260 (0.280)	−0.558 ** (0.269)	−0.493 * (0.271)
W*lnIS				−1.588 (1.634)	−2.102 (1.455)	−0.907 (1.507)
W*ER				0.608 (0.598)	0.650 (0.562)	0.627 (0.564)
constant	28.143 *** (4.761)	22.701 *** (4.824)				

Notes: Standard errors in parentheses; * *p* < 0.10, ** *p* < 0.05, *** *p* < 0.01.

**Table 5 ijerph-18-06111-t005:** The effect decomposition of SPDM.

Variable	Direct Effect		Indirect Effect		Total Effect	
	W1	W2	W1	W2	W1	W2
lnTI	−0.541 *** (0.179)	−0.468 *** (0.177)	−4.599 *** (1.365)	−2.006 (1.298)	−5.140 *** (1.429)	−2.474 * (1.348)
(lnT)^2^	0.049 *** (0.014)	0.049 *** (0.014)	0.460 *** (0.095)	0.247 *** (0.083)	0.509 *** (0.099)	0.297 *** (0.087)
lnP	−0.806 *** (0.281)	−1.043 *** (0.277)	−3.155 (2.419)	−9.633 *** (2.374)	−3.962 (2.547)	−10.676 *** (2.488)
lnA	2.075 *** (0.526)	1.521 *** (0.517)	1.262 (1.591)	0.612 (1.552)	3.337 ** (1.666)	2.134 (1.575)
lnFDI	0.197 ** (0.082)	0.169 ** (0.082)	−0.280 (0.634)	−0.823 (0.620)	−0.083 (0.655)	−0.654 (0.638)
lnIS	−1.598 ** (0.714)	−1.504 ** (0.722)	−5.109 (3.621)	−3.623 (3.379)	−6.707 * (3.811)	−5.127 (3.578)
ER	0.491 *** (0.167)	0.447 *** (0.165)	1.778 (1.309)	1.800 (1.236)	2.269 * (1.379)	2.247 * (1.297)

Notes: Standard errors in parentheses; * *p* < 0.10, ** *p* < 0.05, *** *p* < 0.01.

**Table 6 ijerph-18-06111-t006:** Empirical results of three dimensions of green development.

Variable	Economy (W1)			Environment (W1)			Society (W1)		
	Direct Effect	Indirect Effect	Total Effect	Direct Effect	Indirect Effect	Total Effect	Direct Effect	Indirect Effect	Total Effect
lnTI	−0.391 *** (0.090)	−2.318 *** (0.822)	−2.708 *** (0.858)	0.313 *** (0.096)	−1.101 *** (0.352)	−0.788 ** (0.362)	−0.479 *** (0.100)	−1.068 (0.965)	−1.546 (1.006)
(lnTI)^2^	0.029 *** (0.007)	0.257 *** (0.057)	0.286 *** (0.060)	−0.031 *** (0.008)	0.071 *** (0.025)	0.040 (0.025)	0.050 *** (0.008)	0.137 ** (0.068)	0.188 *** (0.070)
lnP	−0.469 *** (0.143)	−3.873 ** (1.510)	−4.342 *** (1.583)	0.127 (0.146)	2.272 *** (0.652)	2.399 *** (0.681)	−0.461 *** (0.158)	−4.426 ** (1.802)	−4.887 *** (1.884)
lnA	1.687 *** (0.260)	−0.485 (0.964)	1.202 (1.013)	−0.340 (0.295)	1.935 *** (0.468)	1.595 *** (0.439)	0.736 ** (0.287)	−0.526 (1.120)	0.210 (1.178)
lnFDI	0.157 *** (0.041)	−0.294 (0.392)	−0.137 (0.405)	−0.001 (0.045)	0.108 (0.169)	0.107 (0.169)	0.043 (0.046)	−0.020 (0.456)	0.023 (0.471)
lnIS	−0.713 ** (0.356)	−0.226 (2.256)	−0.939 (2.371)	−0.243 (0.393)	−0.169 (0.967)	−0.412 (0.987)	−0.658 * (0.394)	−4.856 * (2.662)	−5.513 ** (2.792)
ER	0.229 *** (0.085)	0.800 (0.818)	1.029 (0.858)	0.139 (0.088)	−0.323 (0.344)	−0.184 (0.361)	0.117 (0.094)	1.519 (0.962)	1.636(1.008)

Notes: Standard errors in parentheses; * *p* < 0.10, ** *p* < 0.05, *** *p* < 0.01.

**Table 7 ijerph-18-06111-t007:** Threshold analysis of TI on green development in the YREB.

Variable	Total	Economy	Environment	Society
lnTI < 9.79 (17,854)	0.352 *** (0.101)			
lnTI ≥ 9.79 (17,854)	0.708 *** (0.108)			
lnTI < 8.84 (6905)		0.055 *** (0.054)		
lnTI ≥ 8.84 (6905)		0.167 *** (0.053)		
LnTI < 5.79 (327)			0.114 ** (0.050)	
5.79 ≤ lnTI ≤ 5.84 (343)			0.682 *** (0.081)	
lnTI ≥ 5.84 (343)			0.159 *** (0.059)	
lnTI < 9.17 (9604)				0.153 *** (0.057)
lnTI ≥ 9.17 (9604)				0.342 *** (0.058)
lnP	−0.809 *** (0.309)	−0.558 *** (0.162)	0.130 (0.150)	−0.570 *** (0.174)
lnA	2.133 *** (0.281)	1.197 *** (0.148)	0.571 *** (0.137)	0.368 ** (0.158)
lnFDI	−0.010 (0.097)	0.046 (0.051)	0.003 (0.047)	−0.053 (0.054)
lnIS	−5.123 *** (0.681)	−2.198 *** (0.359)	−0.328 (0.337)	−2.752 *** (0.384)
ER	0.587 *** (0.187)	0.283 *** (0.098)	0.142 (0.091)	0.200 * (0.106)
Constant	24.067 *** (3.491)	7.469 *** (1.824)	0.078 (1.718)	18.084 *** (1.957)
*Obs.*	1620	1620	1620	1620

Notes: Standard errors in parentheses; * *p* < 0.10, ** *p* < 0.05, *** *p* < 0.01.

## Data Availability

Not applicable.

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
