# Peer review of "Does Technological Innovation Promote Green Development? A Case Study of the Yangtze River Economic Belt in China"

_ijerph, 2021, doi:10.3390/ijerph18116111_

Round 1

Reviewer 1 Report

1 The introduction outlines quite clearly the macro-theme of the role between technological innovation (IT) and sustainable development, even if the international framework in which the specificity of the Chinese context is inserted, is not well defined (few references). Chinese context is instead sufficiently defined.

It is suggested to extend the international framework

2  

In the discussion, the three dimensions considered (economic, social and environmental) are not described in a systematic way but with very concise descriptions; the references to specific situations need to be referred a previous general framework.

3

In paragraph 4.2 it is suggested to define the FE index of table 4, as was done with the SPDM index.
In paragraph 4.2.1 the description of the results (line 348-368) could be systematized and described in a more immediate way by means of a bulleted list, for example.

4

The ref. 33 in the text does not seem consistent.
In the text (line 129) we talk about China but that analysis concerns the US.

check the following sentences
- from line 442 to line 445
- from 465 to line 466 (as incipit of the paragraph). 

Reviewer 2 Report

At first I would like to thank the authors for providing me the opportunity of reading their work and make my comments on the topic. I expect that they find it of use.

  1. Concerning the expression "technological innovation" there is a need for further explanation as it is not evident to which type(s) of conventional innovation they reffer to.
  2. The conceptual model is accurate, however, the authors should consider the simplification in terms of the explanation - I maen, the reader needs to understand why did they build the model as is and what is the literature that supports each connection established. 
  3. In terms of the empirical procedures there is too much going on. The models implemented are very interesting but it is very hard to follow. Please consider a simplification of the empirical procedures. 
  4. I believe that cutting some parts of the estimation and including them as aprrendixes would help the readers.
  5. Please consider some clarifications in the empirical part and brief explanations about the purpose of the instrument as well as the explicit finding. In the section "policy implications" I believe that the paper:  Costa, J. Carrots or Sticks: Which Policies Matter the Most in Sustainable Resource Management? Resources 202110, 12. https://doi.org/10.3390/resources10020012. Should be considered as it provides an insight about the accuracy of the policy actions.

I wish the autohrs the best of luck in their research.

Reviewer 3 Report

I would like to congratulate the authors for the  paper and its empirical contribution. It was a pleasure to participate in this review process. I would like to make a number of recommendations in order to try to contribute to its possible improvement.

Abstracts

I believe that the main results obtained and more concrete conclusions should be presented in a very introductory way.

Introduction

The introduction is clear and sets the structure of the work. It would be interesting to link this type of study not only to China but also to take examples from other geographical areas. Lines 62-71 give an idea of the objectives and methodology of the paper. I suggest that the authors focus more attention on this point so that it is justified and suitably structured from the outset, especially the so-called "three dimensions" (lines 66-69).

Literature review

I believe that the literature review should be more extensive, focusing on the aspects that will subsequently guide the research. It would be important to point out, perhaps in the form of a graph or table, a summary of contributions including economic and numerical aspects. Similarly (lines 128-134), it would be interesting to contrast academic contributions in favour and against this economic dimension in the Green Development concept.

Methodology

It would be interesting to contrast previous academic works (as well as the favourable aspects in their use) that justify the method used in the research (lines 184 - 186). I suggest the authors to raise these questions a priori.

Result, discussion and conclusions

I congratulate the authors for the results analysis part and for the econometric model developed. It would be interesting to emphasise the "Discussion" part and to provide more bibliography and possible limitations to the study, both in the technical, political and economic aspects. The conclusions section (I recommend that the authors do not include references in this part) should be more developed and should highlight the results obtained previously. It seems to me to be too short for the scope of the work.

I wish the authors luck in the revision process, which I am sure will be successful.

Reviewer 4 Report

The paper focuses on an interesting and relevant issue. Technological innovation are of crucial importance in fostering economic growth and green development. The study, however, needs further elaboration and corrections.

The phrase … “ the mechanism of TI on regional green development” in the second sentence of the Abstract should be revised, apparently some words are missing.

In lines 54-55 the Authors argue that “Regions are in different stages of development, which leads to differences in the impact of TI on economic greening, ecological greening, and social greening”. It is not entirely clear however if this statement refers to regions in general or to the Chinese ones.

Lines 93-94 – the word “regulation” does not seem suitable.

Lines 150-151 – the style of the sentence “It is scientific and reasonable to have chosen the YREB as the research area.” needs revision.

As regards the choice of third-class indexes used to estimate the green development index in Table 1. (lines 177-178):

  1. The Authors have chosen “Industrial sulfur dioxide emission intensity (ton per km2 )” as one of the environmental impact indicators, however the majority of environmental regulations worldwide focuses on carbon dioxide emissions. It would be therefore advisable to include the latter measure in the analysis or to explain its lack clearly to the Readers.
  2. The Authors have chosen domestic water consumption per capita and domestic electricity consumption per capita as measures of social impact assuming their positive impact on GDL. It is not clear however, why would higher consumption of water and electricity contribute to green development positively.

Lines 230-236 – the Authors have chosen the number of patent applications as a measure of technological innovation. It would be advisable to use a measure of patent intensity instead (patent applications per capita).

In lines 357-360 the Authors claim that hat: “Population has an inhibiting effect on the improvement of GDL…because population will increase productivity and reduce environmental pollution through demonstration effects.” The reasoning is not clear especially given the assumed negative impact of population growth on GDL in Table 1. and further in lines 242-246.  

Round 2

Reviewer 2 Report

Congratulations to the authors for bringing an improved version of their article. In my view several points were clarified as well as the readability of the document.

However there are some aspects requiring further improvement  - some sentences are too long and misleading, requiring for rephrasing or revision. 

Along the paper there are some typos and the referencing lacks spacing in some cases.

My major concern is related to the model interpretation - Please carefully revise the entire text as there are some incorrect comments: in line 375 the authors have commented the value 0.091% which corresponds to the standard deviation.

Also, the stars attributed by significance levels are incorrect in several cases or the authors comment insignificant coefficients (e.g. 0.142), please revise the tables and perhaps improve the fomatting as it is very difficult to follow the line and identify the coefficient for each variable.

Please provide some paragraphs with the theoretical and the practical implications of your research as well as some limitations of your study.

Best of luck with your research.
